# Phosphorylation of AMPKα at Ser485/491 Is Dependent on Muscle Contraction and Not Muscle-Specific IGF-I Overexpression

**DOI:** 10.3390/ijms241511950

**Published:** 2023-07-26

**Authors:** Chih-Hsuan Chou, Elisabeth R. Barton

**Affiliations:** 1Applied Physiology & Kinesiology, College of Health and Human Performance, University of Florida, Gainesville, FL 32611, USA; cccbw@missouri.edu; 2Biomedical Sciences, College of Veterinary Medicine, University of Missouri, Columbia, MO 65211, USA

**Keywords:** AMPK, skeletal muscle, IGF-I, muscle contraction

## Abstract

Glucose is an important fuel for highly active skeletal muscles. Increased adenosine monophosphate (AMP)/adenosine triphosphate (ATP) ratios during repetitive contractions trigger AMP-activated protein kinase (AMPK), indicated by phosphorylation of AMPKα^Thr172^, which promotes glucose uptake to support heightened energy needs, but it also suppresses anabolic processes. Inhibition of AMPK can occur by protein kinase B (AKT)-mediated phosphorylation of AMPKα^Ser485/491^, releasing its brake on growth. The influence of insulin-like growth factor I (IGF-I) on glucose uptake and its interplay with AMPK activation is not well understood. Thus, the goal of this study was to determine if increased muscle IGF-I altered AMPKα phosphorylation and activity during muscle contraction. Adult male mice harboring the rat *Igf1a* cDNA regulated by the fast myosin light chain promoter (*mIgf1^+/+^*) and wildtype littermates (WT) were used in the study. *mIgf1^+/+^* mice had enhanced glucose tolerance and insulin-stimulated glucose uptake, but similar exercise capacity. Fatiguing stimulations of extensor digitorum longus (EDL) muscles resulted in upregulated AMPKα phosphorylation at both Thr172 and Ser485/491 in WT and *mIgf1^+/+^* muscles. No differences in the phosphorylation response of the downstream AMPK target TBC1D1 were observed, but phosphorylation of raptor was significantly higher only in WT muscles. Further, total raptor content was elevated in *mIgf1^+/+^* muscles. The results show that high muscle IGF-I can enhance glucose uptake under resting conditions; however, in contracting muscle, it is not sufficient to inhibit AMPK activity.

## 1. Introduction

Optimization of skeletal muscle mass and function relies on the regulation of anabolic and metabolic pathways to meet the needs of tissue activity. Insulin-like growth factor I (IGF-I) plays a role in both anabolic and metabolic functions in skeletal muscle. Decreased muscle IGF-I production leads to loss of muscle mass, exercise tolerance, and glucose metabolism [1]. This is one consequence of aging, when diminished circulating IGF-I may induce anabolic resistance for muscle maintenance or muscle growth while deteriorating the action of insulin in glucose uptake [2]. Strategies to boost muscle IGF-I have been used in several animal models, resulting in improved muscle regeneration and functional muscle mass [3,4,5,6,7,8,9]. In addition to enhanced anabolic actions in muscle, there is also metabolic benefit. Specifically, *Igf1a* transgenic mice display increased glucose disposal rate during insulin tolerance tests as well as increased glucose transporter 4 (GLUT4) content [10]. Resistance exercise training has been shown to increase localized muscle IGF-I secretion and, therefore, to induce greater muscle cross-sectional area and enhanced glucose tolerance, and, as such, can serve as an optimal non-invasive intervention in the elderly population and patients with metabolic disorders, such as type 2 diabetes [11,12,13].

Exercise, as a powerful intervention to enhance muscle health, benefits not only physical function but also cellular metabolism. During exercise, adenosine monophosphate (AMP)-activated protein kinase (AMPK) is activated due to increased energy expenditure by muscle contraction, maintaining the balance between anabolic and catabolic programs for cellular homeostasis in response to metabolic stress. The increased AMP/adenosine triphosphate (ATP) ratio within myofibers during repetitive contractions activates phosphorylation of the AMPK α-subunit at Thr172, inducing downstream signaling cascades. This includes the promotion of glucose uptake via regulation of the Rab-GTPase-activating proteins, such as TBC1 domain family member 4 (TBC1D4) (also known as Akt substrate of 160 KDa [AS160]). The phosphorylation levels of TBC1D1 and TBC1D4 ultimately cause GLUT4 translocation [14]. In addition to Thr172, Ser485/491 is another AMPKα epitope that can be phosphorylated [15]. While phospho-AMPKα^Thr172^ is well-studied and enhances AMPKα activity, the phosphorylation site at Ser^485/491^ was identified in the past decade and is thought to inhibit AMPKα activity [16,17]. 

Recent studies demonstrated that the activation of the protein kinase B (AKT) pathway triggers AMPKα Ser485/491 phosphorylation in cultured myotubes and in ex vivo muscles after resistance exercise [17,18]. AKT phosphorylation was achieved by insulin stimulation, but the same effect was observed in myotubes following administration of exogenous IGF-I. This suggests that IGF-I may be an indirect modulator of AMPKα activity via AKT activation in skeletal muscle. However, to our knowledge, no study has demonstrated an interconnection between IGF-I on AMPKα regulation in skeletal muscle after muscle contraction. Whether local muscle IGF-I plays a role in phosphorylating AMPKα^Ser485/491^ and how it influences the exercise response in unknown. The aim of this study was to determine if increased muscle IGF-I altered AMPKα^Ser485/491^ and AMPKα^Thr172^ phosphorylation and activity at rest or during muscle contraction. 

## 2. Results

We used 16-to-20-week-old male mice with and without the rat *Igf1a* transgene (*mIgf1^+/+^*) to investigate the influence of increased muscle IGF-I on AMPK. We first profiled body composition and muscle IGF-I content to validate the phenotype of *mIgf1^+/+^* mice. *mIgf1^+/+^* mice had 20- to 60-fold higher IGF-I protein in tibialis anterior muscles than wildtype (WT) littermates, which was confirmed by IGF-I ELISA (Figure 1A). Consistent to previous studies, *mIgf1^+/+^* mice had higher body weight compared to their WT littermates (Figure 1B). The absolute lean mass in grams was similarly affected, in which *mIgf1^+/+^* mice had increased lean mass compared to their control groups (Figure 1C). There was no difference in absolute fat mass observed between groups (Figure 1D). Furthermore, there were no apparent differences in normalized fat or lean mass that were dependent on muscle IGF-I overexpression (Figure 1E,F). We examined the metabolic characteristics of mice using glucose (GTT) and insulin (ITT) tolerance tests. *mIgf1^+/+^* mice had enhanced glucose clearance in GTT (Figure 2A,B). The increased glucose uptake after insulin induction was also observed under ITT (Figure 2C,D). Circulating insulin levels in fasted animals were not different between genotypes (mean ± SD, 1.72 ± 0.61 vs. 2.18 ± 0.58 ng/mL, WT vs. *mIgf1^+/+^*, N = 4 and 8; unpaired *t*-test, *p* = 0.23). Our results support the phenotypes observed in previous studies, in which increased muscle IGF-I content results in increased lean mass and body weight and improved glucose metabolism [6,10]. 

To investigate the exercise capacity of *mIgf1^+/+^* mice, we used a run-to-exhaustion (R2E) test. The total running distance showed no statistical difference between WT vs. *mIgf1^+/+^* mice (mean ± SD, 355.2 ± 76.46 vs. 389.2 ± 151.4 m, WT vs. *mIgf1^+/+^*, N = 13–18; unpaired *t*-test, *p* = 0.4634) (Figure 3A). The maximal speed of R2E at termination is illustrated in Figure 3B using a violin plot. Maximal speed in WT mice ranged from 24 to 33 m/min while *mIgf1^+/+^* mice had a wider range from 21 to 39 m/min (mean ± SD, 27.00 ± 2.739 vs. 28.00 ± 5.247 m/min, WT vs. *mIgf1^+/+^*). The mode of maximal running speed in WT was 27 m/min (6 out of 13) while 24 m/min was observed in *mIgf1^+/+^* (5 out of 18). There was no significance in either total running distance or maximal speed between strains.

We used isolated EDL muscles to measure force generation and invoke activation of AMPKα. At baseline, *mIgf1^+/+^* mice had significantly higher EDL muscle mass (mean ± SD, 10.40 ± 0.917 vs. 14.91 ± 1.037 mg, WT vs. *mIgf1^+/+^*, N = 6 and 10; unpaired *t*-test, *p* < 0.0001) and tetanic force (mean ± SD, 355.4 ± 34.2 vs. 434.4 ± 32.9 mN, N = 6 and 10; unpaired *t*-test, *p* = 0.0004) than WT (Figure 3C,D). However, after normalizing by cross-sectional area, there was no significant difference between WT and *mIgf1^+/+^* EDL specific force (Figure 3E). An 8 min bout of stimulation was used to generate fatigue in the muscles. Force production profiles during the fatigue protocols showed similar force drops between WT and *mIgf1^+/+^* EDLs (Figure 3F). Thus, all measures of function were comparable between strains when normalized for the increased EDL muscle mass in *mIgf1^+/+^* mice.

Immunoblotting was used to quantify AMPK phosphorylation at Thr172 and S485/491 in resting EDL (UNSTIM) and the contralateral EDL from the same animal after fatiguing stimulation (STIM) (Figure 4A). In WT and *mIgf1^+/+^* muscles, there was a main effect of stimulation (two-way ANOVA, *p* < 0.05), and a significant upregulation of phospho-AMPKα^Thr172^ after electrical stimulation in both WT and *mIgf1^+/+^* EDL muscles (STIM vs. UNSTIM) by post hoc analysis (Figure 4B). Similarly, phospho-AMPKα^Ser485/491^ showed a stimulation-dependent increase in EDLs from both strains of mice (two-way ANOVA and post hoc, *p* < 0.05) (Figure 4C). Neither epitope displayed different phosphorylated or total AMPKα levels depending on the strain in the resting muscle. Furthermore, the comparison of the effect of the fatiguing stimulation by calculating the fold change of phosphorylation revealed no statistical differences between strains (Figure 4D,E). Taken together, the fatiguing stimulation enhanced phosphorylation of AMPKα^Thr172^ and AMPKα^Ser485/491^, and there was no enhancement of phospho-AMPKα^Ser485/491^ caused by increased IGF-I. 

Given that phosphorylation of AMPKα^Ser485/491^ is a substrate of AKT [16,17], we further tested the phosphorylation of AKT1 at Ser473, which is hypothesized to be the upstream kinase that phosphorylates AMPKα^Ser485/491^. Both AKT1 and AKT2 isoforms were apparent in the blots for total AKT (Figure 5A); however, the phosphorylation responses were observed only in AKT1, and were used to assess differences in phosphorylation levels. We found that EDL muscles from WT as well as *mIgf1^+/+^* mice exhibited significantly increased phospho-AKT1^Ser473^ by muscle stimulation (two-way ANOVA, main effect in stimulation and post hoc UNSTIM vs. STIM, *p* < 0.05) (Figure 5B). On the other hand, phospho-AKT1^Thr308^, which is considered to be the downstream target of IGF-I signaling, was also upregulated after repeated muscle contractions in both WT and *mIgf1^+/+^* EDLs (Figure 5A,C). No significance was found for phosphorylation responses at either Ser473 or Thr308 between strains, nor was there any difference in phosphorylation state between strains in resting conditions (Figure 5B–E). Overall, AKT1 phosphorylation was dependent upon stimulation status and not on muscle IGF-I levels. 

AMPK increases glucose uptake by mediating GLUT4 translocation via deactivation of TBC1D1 [14]. Phosphorylation of Ser700 is the TBC1D1 target of active AMPK [19], and this residue served as an indication of downstream activity by AMPK. Basal phospho-TBC1D1^Ser700^ was not different between strains, yet, after fatiguing stimulation, EDL muscles from *mIgf1^+/+^* mice had elevated phosphorylation levels at this residue (Figure 6A,B). This suggests that the potential inhibition of AMPK by high IGF-I was insufficient to block AMPK-mediated phosphorylation of TBC1D1^Ser700^.

In addition to mobilizing fuel when ATP is low, AMPK also suppresses anabolic processes, in part through rapamycin-sensitive mTOR (mTORC1) pathway inhibition via the phosphorylation of the mTOR binding partner raptor [20]. We measured the phosphorylation of raptor at Ser792, the target of AMPK activity, and found that both stimulation and strain were significant by two-way ANOVA (*p* = 0.01, and 0.004, respectively). EDLs from WT mice exhibited a significant response to stimulation (Figure 7A,B), even though the fold change of Ser792 phosphorylation was evident in both strains (Figure 7C). This appeared to be due to diminished proportional phosphorylation of raptor in the EDLs from *mIgf1^+/+^* mice. Furthermore, the total raptor content was elevated in the EDLs from *mIgf1^+/+^* mice (Figure 7D), in contrast to other proteins examined. Taken together, dampened phosphorylation of raptor combined with the increased content of the protein suggests a strategy by which muscles with high IGF-I content may still retain anabolic drive in the face of high AMPK activity. 

## 3. Discussion

The interactions between anabolic and metabolic pathways are most apparent when they are challenged by changes in activity, glucose abundance, or pathology. In this study, we examined the effects of a predominantly anabolic driver, high IGF-I, on glucose handling and exercise capacity, with a focus on its modulation of a key nutrient and activity sensor, AMPK. While heightened muscle IGF-I led to a ~40% increase in muscle mass as previously shown [6], there was only a small enhancement in glucose clearance and insulin sensitivity, and no difference in exercise capacity compared to littermate controls. The increased muscle mass serves as a sink for bulk glucose clearance. Given that the amount of glucose delivered was normalized to body weight, it supports the idea that the muscles from *mIgf1^+/+^* and WT mice have equivalent abilities to take up glucose. By extension, the improved insulin-stimulated glucose clearance in *mIgf1^+/+^* mice suggests that the increased IGF-I in the muscle may enhance insulin actions through their related receptor pathways. The integrated area under the curves of GTT represents the total glucose clearance from the circulation throughout GTT, which was significantly increased in *mIgf1^+/+^* compared to WT mice. However, we were not able to detect differences between strains at any given timepoint. The variations across individual animals are high and may result from the complexity of hormones involved in glucose uptake during testing. A similar situation was found in the ITT results. We did not collect muscle tissues after GTT or ITT. Therefore, we cannot confirm that the changes observed are due solely to actions in skeletal muscle by IGF-I. More assays, such as uptake measurements of 2-DG glucose [21] or isotopically labeled substrates [22], are needed to validate our findings. From the perspective of exercise capacity, one would predict that an enhancement in glucose uptake afforded by IGF-I would lead to greater running speeds or exercise intensities, but this was not the case. Instead, it appeared that a similar “setpoint” for metabolic needs was achieved in both mouse strains, potentially because different IGF-I levels are present from birth. This contrasts with our previous study in which muscle *Igf1* was deleted in young adult mice and led to impaired glucose clearance following treadmill running [1]. To our knowledge, there were no reports of exercise capacity in *mIgf1^+/+^* mice, motivating us to examine the effects of high IGF-I on the ability of the mice to run. Our results present only one aspect of exercise capability, namely maximal exercise capacity. Future studies performing constant low-intensity running may be needed to investigate the impact of high muscle IGF-I on endurance exercise. 

To eliminate the effects of the organism on IGF-I effects, we utilized isolated muscles and fatiguing contractions to mimic intense exercise, focusing on phosphorylation of AMPKα^Thr172^ as an indicator of exercise intensity. After ex vivo stimulation, the phosphorylation of AMPKα^Thr172^ significantly increased in both WT and *mIgf1^+/+^*. The contraction also induced upregulated phospho-AMPKα^S485/491^ in *mIgf1^+/+^* muscles, supporting the previous observations of IGF-I modulation of this residue via AKT [17,18], but phosphorylation of these sites was also apparent in WT muscles, suggesting that normal levels of IGF-I are sufficient to upregulate phosphorylation of Ser485 as well as phospho-AKT1 at both Ser473 and Thr308 after electrical stimulated muscle contraction. Along these same lines, TBC1D1^Ser700^, a downstream substrate for AMPK, revealed higher phosphorylation after contractions in the EDL muscles from *mIgf1^+/+^* mice than WT mice, an indication that AMPK activity was not inhibited by high IGF-I. Taken together, these results support the idea that contraction-induced AMPK activation is independent from the content of IGF-I in skeletal muscle.

TBC1D1 is one of many AMPK signaling substrates. Phosphorylation of TBC1D1 is thought to be related to GLUT4 translocation after muscle contraction [19,23,24]. TBC1D1 contains multiple phosphorylation sites and has been reported to be induced by different stimulus sources, such as insulin, IGF-I, and muscle contraction [19,25]. Thus, the patterns of TBC1D1 phosphorylation may provide insights into which upstream factor is driving phosphorylation and modulating GLUT4 translocation via TBC1D1 actions [26]. We chose phospho-TBC1D1^Ser700^ to investigate the activation of AMPK because this residue is uniquely sensitive of AMPK after muscle contraction [27]. Significantly elevated phospho-TBC1D1 after electrical stimulation was seen only in *mIgf1^+/+^* muscles. However, we cannot discount that the phosphorylation levels may differ at later timepoints following fatiguing contractions. We collected stimulated EDLs within 20 min after the 8 min fatigue test was completed. Previous studies showed that the influence of exercise on GLUT4 via TBC1D1 occurred during recovery instead of immediately after exercise [24,28]. This may explain the variation in phospho-TBC1D1 observed in our study, in which the duration between muscle contraction completion and tissue processing may not have been long enough to induce the downstream signaling cascades. The downstream responses of AMPKα activation include not only glucose uptake but can also increase lipid oxidation [29,30]. Further studies may be needed to address phospho-TBC1D1 at various timepoints to understand muscle IGF-I in responding to exercise regarding AMPK and TBC1D1 in depth. Other downstream targets associated with fuel utilization, such as acetyl-co A carboxylase, may also be needed to investigate the pathways activated after muscle contraction in different muscle IGF-I contents. 

In contrast to the above scenario, a second downstream AMPK target, raptor^Ser792^, displayed increased phosphorylation only in stimulated WT muscles. However, this was also associated with higher raptor content in *mIgf1^+/+^* muscles. This presents a different mechanism by which muscles with high IGF-I may escape the suppression of growth by AMPK activity. Raptor is a key protein in mTORC1 anabolic activity, as its absence leads to a myopathic phenotype [31]. Previous studies have shown that raptor association with mTOR is phosphorylation-independent, yet its association with inhibitory factors, such as 14-3-3, are increased with phosphorylation, thereby recruiting 14-3-3 to mTOR and shutting down biosynthesis [20]. Rather than dynamically tuning AMPK through phosphorylation of an inhibitory site, the muscles may have adapted by increasing the amount of raptor. Thus, the bulk increase in raptor populates mTOR binding, prevents its association with phospho-raptor, and allows pro-growth actions to continue. A similar scenario has been described in human cancer cell lines, where resistance to PI3K-mTOR inhibition was linked to elevated raptor [32]. This possibility was not directly tested in our study, but future experiments could delve further into this potential mechanism.

Skeletal muscle is able to store IGF-I locally, and the *mIgf1^+/+^* transgenic muscles have at least 20-fold more IGF-I than those from WT mice. It is clear that the growth factor is effective in driving muscle hypertrophy, but whether the reservoir is mobilized for activating IGF-I receptors by acute contractile activity is not clear. Examination of the upstream kinase phosphorylation, AKT1^Ser473^, showed that both WT and *mIgf1^+/+^* transgenic muscles had significantly increased phospho-AKT1^Ser473^ after stimulation, and the level of IGF-I did not influence the level of phosphorylation at either baseline or after stimulation. A similar upregulated response after muscle contraction was observed at phospho-AKT1^Thr308^, which is the epitope considered more sensitive to IGF-I receptor activation, indicating that a high level of muscle IGF-I does not alter the contraction-induced phosphorylation at either Ser473 or Thr308 of AKT1. The previous observations of a regulatory link between IGF-I and AMPKα were established with exogenous IGF-I applied to cultured myotubes [17], but, in the current study, it was challenging to show similar effects in isolated muscles. Indeed, previous studies compared the IGF-I levels following acute bouts of resistance or aerobic activity, and increased IGF-I protein was evident 1 h after activity, but only in resistance exercise [33]. Furthermore, this increase may have entered the tissue from the circulation, where IGF-I is also abundant [1]. In our study, we used isolated muscles to eliminate the contribution of circulating IGF-I. We were not able to pursue measurements of IGF-I receptor phosphorylation in the same EDL due to the limited protein available, nor did we examine the transgenic muscles for release of the active IGF-I ligand [34]. Future measurements could address these potential reasons for the modest modulation of AMPK activity in *mIgf1^+/+^* muscles.

While AMPK activation is generally accepted as an exercise-responsive pathway to enhance glucose uptake and lipid metabolism during exercise, the role of AMPK may be more important after exercise for recovery instead of during exercise [35]. The upregulated AMPK activities may be mainly due to the increase in intracellular stress but not the demand for ATP in prolonged exercise or muscle stimulation. In our ex vivo muscle contraction experiment, although isolated EDLs were incubated in an oxygenated and nutrient-rich environment, we could not avoid the possible hypoxia induced by the high frequency and repetitive muscle contractions, which could contribute to AMPK activation as well. 

In summary, our study supports that lifelong high muscle IGF-I enhances glucose handling, but there is modest modulation of AMPK activity. Instead, high levels of IGF-I in muscle may counter AMPK-mediated suppression of growth by enabling mTOR activity to persist. 

## 4. Materials and Methods

### 4.1. Animals

All procedures were approved by the Institutional Animal Care and Use Committee of the University of Florida. Male transgenic mice harboring the rat *Igf1a* cDNA under the control of the fast myosin light chain promoter–enhancer (*mIgf1^+/+^,* MLC/mIgf), resulting in enriched expression of *Igf1* restricted to skeletal muscle [6], were used in the study to examine *Igf1* overexpression in skeletal muscle. Mice were backcrossed onto the C57Bl6 strain for 10 generations prior to their use in this study. Age- and sex-matched littermates without the transgene were used as wildtype controls (WT). All mice were 16–20 weeks old at the time of the experiments. Genotypes were confirmed by endpoint polymerase chain reaction (PCR) on genomic DNA samples with primers specific for the transgene (sense: 5′-tgctcacctttaccagctcgg-3′; antisense: 5′-gcccggatggaacgagctgact-3′).

### 4.2. Body Composition

Mice were subjected to time domain (TD)-NMR (LF90 Minispec, Bruker, Spring, TX, USA) to determine body compositions of fat, lean mass, and fluid in conscious live mice under randomly fed conditions. The mice were placed into a Plexiglas sample holder (90 mm in diameter and 250 mm in length), with ventilation holes provided at both ends and around the tube circumference, and then inserted into the 0.5 T magnet bore. The time of measurement for each reading was approximately 1 min in duration, and all measurements were performed in triplicate, with the average reported. After the measurements were obtained, the mice were returned to their cages.

### 4.3. Glucose Tolerance Test (GTT)

We used GTT to investigate the capacity of glucose clearance via peripheral tissues in our mouse models. Mice were subjected to overnight (5 p.m. to 9 a.m.) fasting before GTT. Tests were performed at 9 a.m. and 2 g/kg bodyweight of glucose was delivered to animals via intraperitoneal injection. Glucose levels (mg/dL) were measured at 0, 15, 30, 60, 90, and 120 min after glucose injection by a tail blood sample using a glucometer (ReliOn Prime Glucometer; Wal-Mart Stores Inc., Bentoville, AR, USA). Following the tests, mice were returned to their home cages with ad lib chow.

### 4.4. Insulin Tolerance Test (ITT)

ITT was utilized to measure temporal glucose uptake after insulin injection to examine the sensitivity of the mice to insulin actions. Mice were fasted for 4 h prior to ITT and 1.0 U/kg lean-mass insulin was injected intraperitoneally. Glucose levels (mg/dL) at 0, 15, 30, 60, and 90 min after injection were measured via a tail blood sample using a glucometer mentioned above. Following the tests, mice were returned to their home cages with ad lib chow. ITT was performed at least 48 h after GTT was completed.

### 4.5. Run-to-Exhaustion (R2E) Tests

Treadmill running was utilized to evaluate the endurance capacity of the mice. Mice were familiarized with treadmill (EXER4/4, Columbus Instruments, Columbus, OH, USA) including a 5 min rest on the treadmill followed by a 5 min run at 18 m/min 2 days before the R2E test. On the test day, the treadmill speed started at 6 m/min, and 3 m/min was added every 3 min until the mouse failed to catch up with the speed. Exhaustion was defined by the mouse falling to the end of the treadmill belt 30 times. Total running time was measured, and the total running distance was calculated for further analysis.

### 4.6. Ex Vivo Muscle Contraction

Isolated muscle mechanics were performed to evoke changes in AMPK phosphorylation. Randomly fed mice were anesthetized with ketamine and xylazine. Both extensor digitorum longus (EDL) muscles were carefully dissected following the protocol in previous studies [36]. After EDLs were dissected, mice were euthanized by cervical dislocation. Additional tissues were harvested and snap-frozen in liquid nitrogen for validating *Igf1* transcripts and IGF-I content in each mouse. One EDL muscle was utilized for stimulation (STIM) while the other muscle rested in the oxygenated Ringer’s solution held at 22 °C for 30 min (UNSTIM). For the STIM muscle, single twitch stimulation with gradual lengthening of the muscle was used to determine optimum length (*L_o_*). All testing protocols were performed with muscles at *L_o_*. In a pilot study, 3 different stimulation patterns were used to maximize the contraction-induced response of AMPK phosphorylation: 40 Hz at 1 s/min for 30 min, 40 Hz at 500 ms/min for 30 min, or 40 Hz at 330 ms/s for 8 min (fatigue test). Muscles exhibited the greatest phosphorylation response following the fatigue test, and this was adopted for all subsequent experiments. After stimulation, STIM and UNSTIM muscles were snap-frozen in liquid nitrogen and stored in −80 °C for subsequent biochemical measurements.

### 4.7. IGF-I ELISA

Quantification of muscle IGF-I content was performed using a commercial ELISA kit (R&D systems, Minneapolis, MN, USA; Cat#MG100) and following the manufacturer’s instructions. Frozen tibialis anterior (TA) muscles were weighed and then ground with a mortar and pestle before the resultant powder was submerged in 150 μL of PBS and frozen overnight at −80 °C. Samples were then centrifuged at 12,000 rpm at 4 °C, and the supernatant was utilized for measurements. A total of 50 μL of each sample was added to each well in duplicate in a 96-well plate. An aliquot of the supernatant was also used for bicinchoninic acid (BCA) assay (Thermo Scientific, Waltham, MA, USA; Cat#23225) to quantify total protein content following the manufacturer’s instructions. Measured IGF-I concentration was normalized to total protein content to quantify the amount of IGF-I in each muscle.

### 4.8. Insulin ELISA

Quantification of serum insulin was performed on samples from 4 h fasted mice fasted using a commercial ELISA kit (EZRMI-13K; Millipore Sigma, Burlington, MA, USA), and were measured according to the manufacturer’s instructions and as described in our previous studies [1].

### 4.9. Immunoblotting

STIM and UNSTIM EDLs were ground using a pestle and mortar on dry ice and homogenized in RIPA buffer (Cell Signaling Technology, Danvers, MA, USA, Cat#9806) to isolate proteins. Protein yields were quantified using a BCA assay, as mentioned above. Electrophoresis was performed using Bis–Tris SDS PAGE (4–12%) gels (Thermo Scientific, WG1402A) with MOPS buffer, and proteins were transferred to nitrocellulose membranes with transfer buffer (Thermo Scientific, Waltham, MA, USA, NP00061) containing 20% methanol under 60 V for 90 min. Membranes were blocked with 5% bovine serum albumin for 1 h at room temperature and further incubated with primary antibodies overnight at 4 °C. Membranes were washed with tris-buffered saline (TBS) containing 0.05% Tween-20 (TBS-T) 3 times and incubated with secondary antibodies with fluorescence at either 680 or 800 nm for 1 h at room temperature. The Li-COR system and software were used for fluorescence imaging and analysis. Antibodies used in the study are listed in Table 1.

### 4.10. Data Analysis

GraphPad Prism (GraphPad version 8.3.0, La Jolla, CA, USA) was used for all data analysis. A normality test (Shapiro–Wilk test) was performed to check whether data met the normal distribution. Student *t*-tests were performed to compare *mIgf1^+/+^* to their WT control counterparts. Two-way repeated ANOVA was used to analyze the interaction between genotype and stimulation (STIM vs. UNSTIM). Sidak’s multiple comparison tests for post hoc analysis were performed if any interaction or main effect was found. The alpha level was set at 0.05.

## Figures and Tables

**Figure 1 ijms-24-11950-f001:**
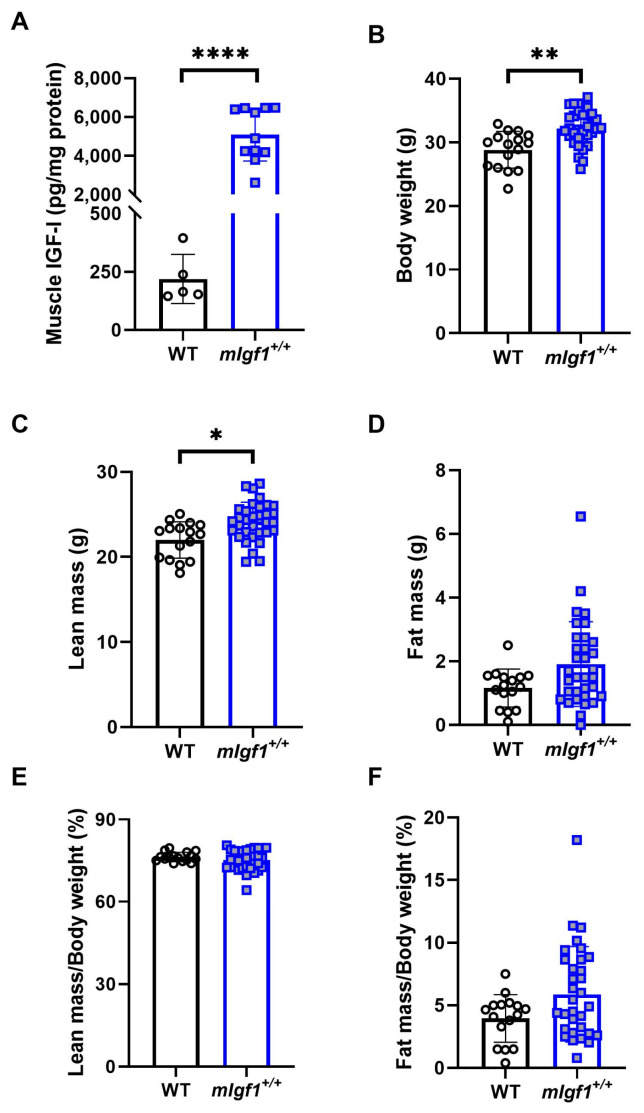
Muscle insulin-like growth factor I (IGF-I) content and body composition in wildtype littermates (WT) and *mIgf1^+/+^* mice. (**A**) IGF-I concentration in TA muscles normalized with total protein content; (**B**) body weight; (**C**) lean mass; (**D**) fat mass; (**E**) percentage lean mass of body weight; (**F**) percentage fat mass of body weight in WT and *mIgf1^+/+^* male mice. N = 5 and 11 for muscle IGF-I content. N = 16 and 33 for body composition measurements. Data presented as individual values and mean ± SD for each group. Independent *t*-tests were performed for each variable between groups. *: *p* < 0.05, **: *p* < 0.005, ****: *p* < 0.0001.

**Figure 2 ijms-24-11950-f002:**
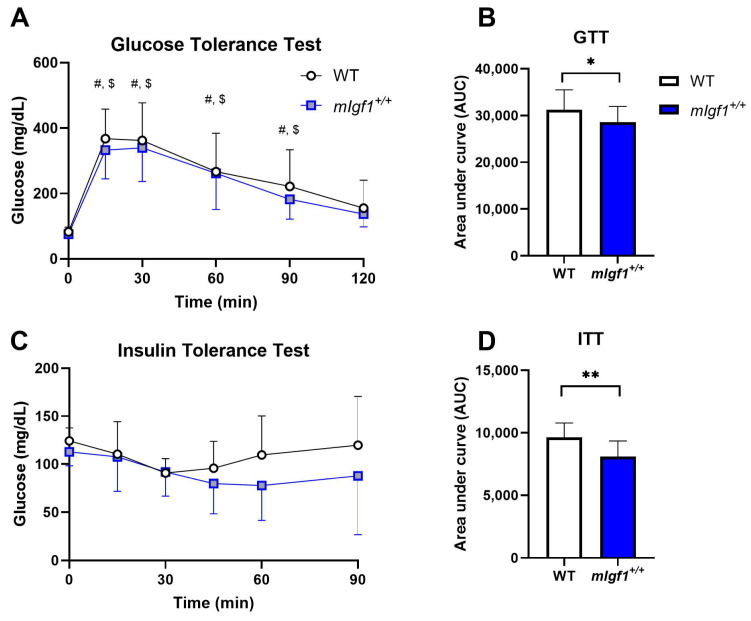
Glucose tolerance test (GTT) and insulin tolerance test (ITT) responses in WT and *mIgf1^+/+^* mice. (**A**) Glucose responses during GTT; (**B**) area under curve analysis for GTT; (**C**) glucose responses during ITT; (**D**) area under curve analysis for ITT. N = 12 and 17 for WT and *mIgf1^+/+^* mice. Data presented as mean ± SD. ^#^: Significance within WT compared to baseline (time 0), *p* < 0.05; ^$^: Significant within *mIgf1^+/+^* compared to baseline (time 0), *p* < 0.05; *: Significance between WT and *mIgf1^+/+^*, *p* < 0.05; **: Significance between WT and *mIgf1^+/+^*, *p* < 0.01.

**Figure 3 ijms-24-11950-f003:**
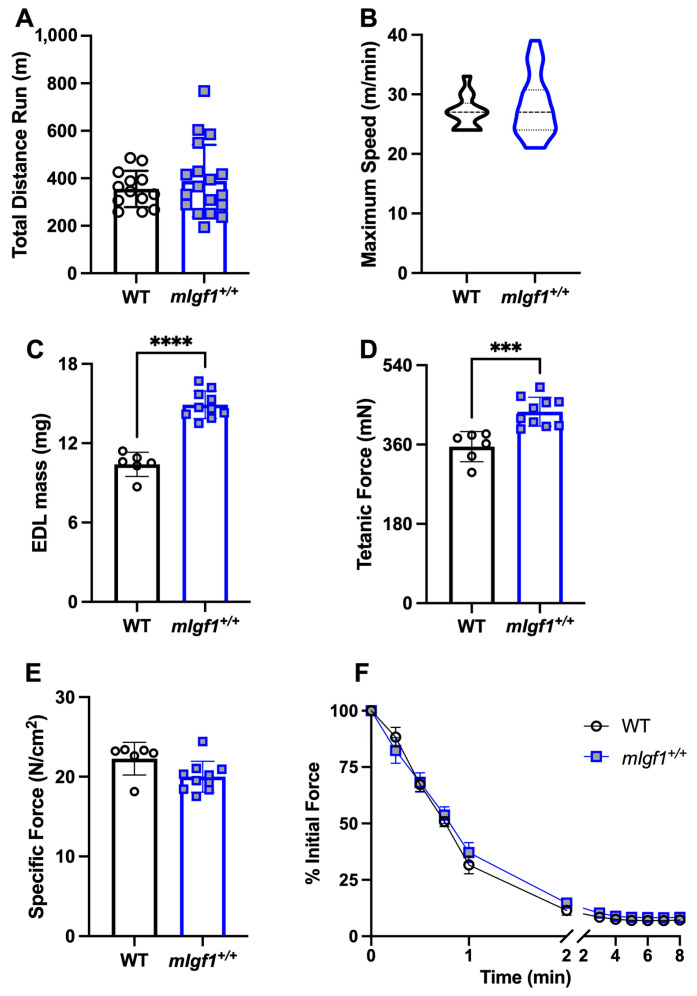
Exercise capacity and extensor digitorum longus (EDL) muscle mechanics. (**A**) Total running distance in run-to-exhaustion (R2E) test; (**B**) maximal running speed in R2E test (data were presented with a violin plot); (**C**) EDL mass in mechanic testing; (**D**) EDL tetanic force; (**E**) EDL specific force; (**F**) EDL force production during fatigue test in WT and transgenic mice. N = 13 and 18 for R2E test. N = 6 and 10 for EDL mechanics. Data presented as individual values and mean ± SD for each group. Independent *t*-tests were performed for each variable between groups. ***: *p* < 0.001, ****: *p* < 0.0001.

**Figure 4 ijms-24-11950-f004:**
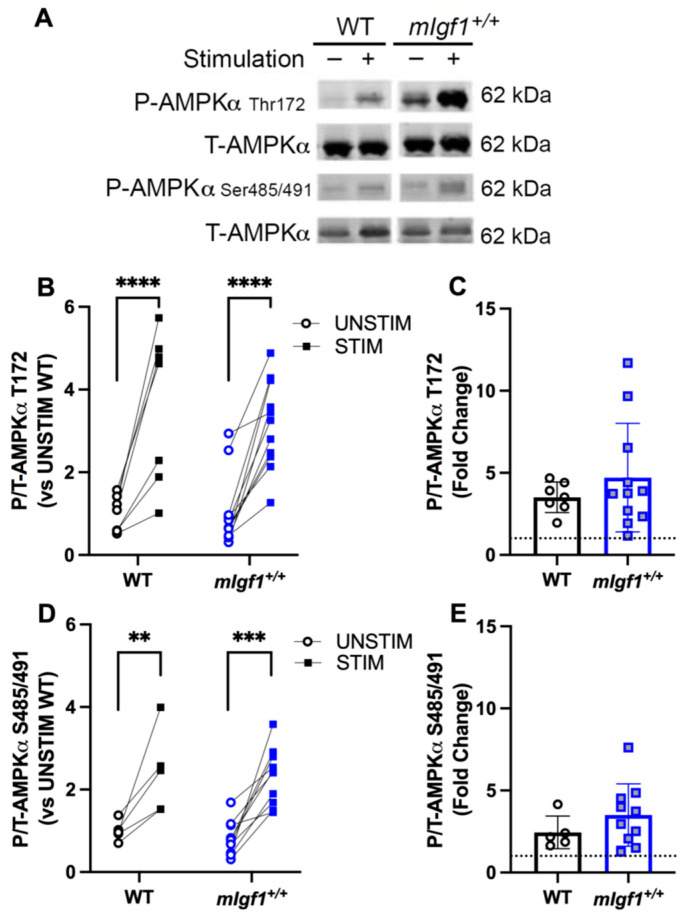
Response of phospho-AMPKα at Thr172 and Ser485/491 after electrical stimulation. (**A**) Representative imaging of western blotting; (**B**) responses of paired EDLs at phospho-AMPKα^Thr172^ for UNSTIM and STIM; (**C**) responses of paired EDLs at phospho-AMPKα^Ser485/491^ for UNSTIM and STIM; (**D**) fold changes from STIM to UNSTIM in phospho-AMPKα^Thr172^; (**E**) fold changes from STIM to UNSTIM in phospho-AMPKα^Ser485/491^. N = 7 and 11 for Thr172 analysis and N = 5 and 11 for Ser485. Data presented as individual values and mean ± SD for each group. **: *p* < 0.005; ***: *p* < 0.0005; ****: *p* < 0.0001.

**Figure 5 ijms-24-11950-f005:**
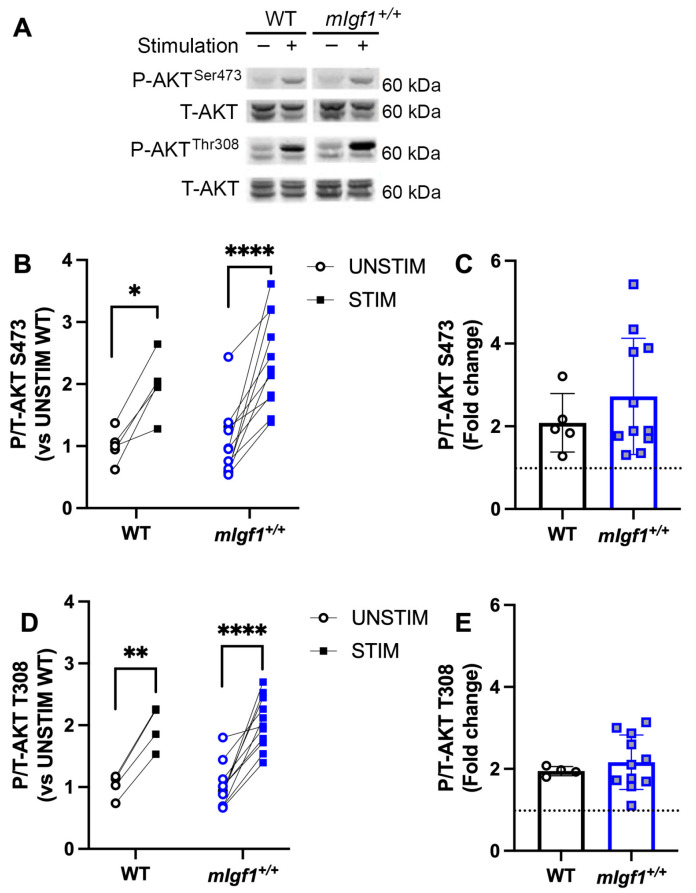
Response of phospho-AKT at Ser473 and Thr308 after electrical stimulation. (**A**) Representative imaging of western blotting; (**B**) responses of paired EDLs at phospho-AKT^Ser473^ for UNSTIM and STIM; (**C**) responses of paired EDLs at phospho-AKT^Thr308^ for UNSTIM and STIM; (**D**) fold changes from STIM to UNSTIM in phospho-AKT^Ser473^; (**E**) fold changes from STIM to UNSTIM in phospho-AKT^Thr308^. N = 5 and 11 for Ser473 analysis and N = 4 and 11 for Thr308. Data presented as individual values and mean ± SD for each group. Significance following post hoc analysis for STIM vs. UNSTIM within group was denoted with asterisks (*). *: *p* < 0.05, **: *p* < 0.005, ****: *p* < 0.0001.

**Figure 6 ijms-24-11950-f006:**
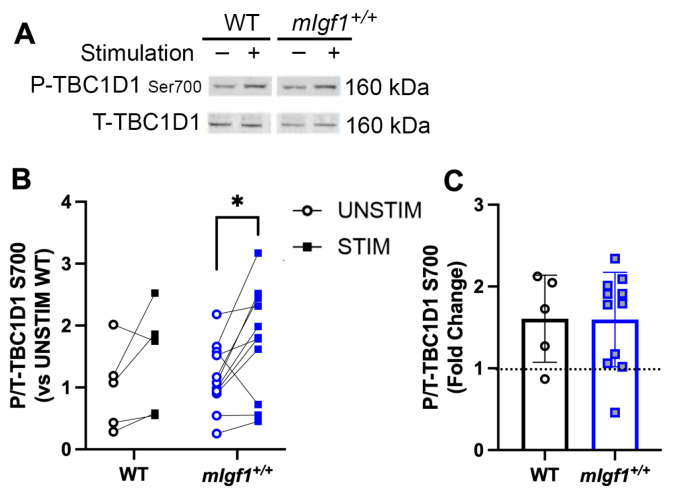
Response of phospho-TBC1D1 at Ser700 after electrical stimulation. (**A**) Representative imaging of western blotting; (**B**) responses of paired EDLs at phospho-TBC1D1^Ser700^ for UNSTIM and STIM; (**C**) fold changes from STIM to UNSTIM in phospho-TBC1D1^Ser700^. N = 5 and 11. Data presented as individual values and mean ± SD for each group. Significance following post hoc analysis for STIM vs. UNSTIM within group was denoted with asterisks (*). *: *p* < 0.05.

**Figure 7 ijms-24-11950-f007:**
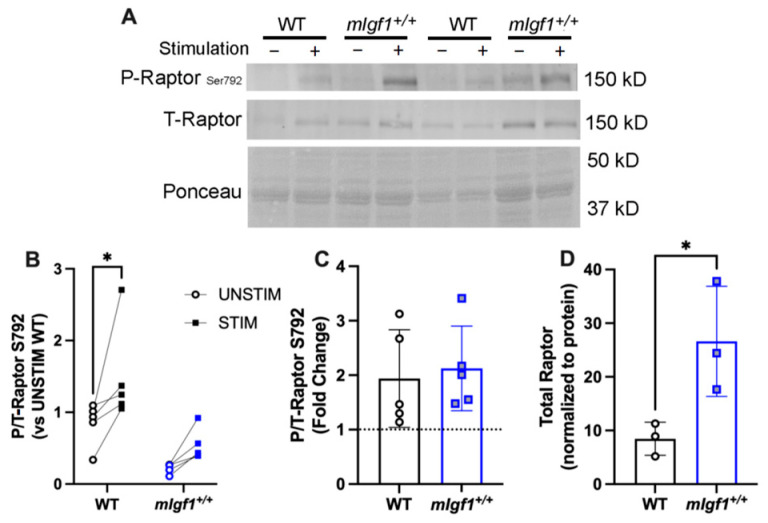
Response of phospho-raptor at Ser792 after electrical stimulation. (**A**) Representative imaging of western blotting; (**B**) responses of paired EDLs at phospho-raptor^Ser792^ for UNSTIM and STIM. Significance denoted for post hoc analysis of STIM vs. UNSTIM within group. (**C**) Fold changes from STIM to UNSTIM in phospho-raptor^Ser792^. N = 5 per strain. (**D**) Total raptor muscle content (N = 3 per strain). Data presented as individual values and mean ± SD for each group. *: *p* < 0.05 for comparisons by unpaired *t*-tests.

**Table 1 ijms-24-11950-t001:** List of antibodies for immunoblotting.

Target	Dilution	Supplier
Total AMPKα	1:2000	Cell Signaling Technology, #2793
Phospho- AMPKα^Thr172^	1:1000	Cell Signaling Technology, #2535
Phospho- AMPKα^Ser485/491^	1:1000	Cell Signaling Technology, #4185
Total AKT	1:2000	Cell Signaling Technology, #2920
Phospho-AKT^Thr308^	1:1000	Cell Signaling Technology, #4056
Phospho-AKT^Ser473^	1:1000	Cell Signaling Technology, #4060
Total TBC1D1	1:1000	Cell Signaling Technology, #66433
Phospho-TBC1D1^Ser700^	1:1000	Cell Signaling Technology, #6929
Total raptor	1:1000	Cell Signaling Technology, #2280
Phospho-raptor^Ser792^	1:1000	Cell Signaling Technology, #2083
Goat-anti-mouse 800 CW	1:15,000	Li-COR
Donkey-anti-rabbit 680 RD	1:15,000	Li-COR

## Data Availability

Not applicable.

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
