# Peer review of "Phosphorylation of AMPKα at Ser485/491 Is Dependent on Muscle Contraction and Not Muscle-Specific IGF-I Overexpression"

_ijms, 2023, doi:10.3390/ijms241511950_

Round 1
Reviewer 1 Report
This manuscript from Chou and Barton examines the effect of genetic IGF-1 overexpression in skeletal muscle on muscle function and signaling pathways with an emphasis on AMP-activated protein kinase (AMPK). The authors conclude that the phosphorylation of IGF-1 after muscle contraction is dependent on IGF-1 overexpression. Although this should, theoretically, work to decrease AMPK activity, no indications of such a deficit were observed. The manuscript is generally well-written, with a straightforward experimental design, but I have several comments and concerns regarding its suitability for publication, as listed below.
Major concerns:
1) I suspect that many of the experiments reported here are underpowered. For instance, in Fig 3E, the distribution of data here strongly suggests that the number of WT animals may be insufficient to detect a difference in specific force between genotypes. Importantly, in Fig 4C: this is a critical experiment because most of the paper’s conclusions derive from it. The phosphorylation of AMPK at S485 is only statistically increased by contraction in the IGF-1 overexpressor group. However, this appears, based on the data distribution, to be due to one seeming outlier in the mIgf1 group. Looking at the WT group, there are only five mice, and all of them increase their phosphorylation status. It is likely that with an increased WT n-size, the conclusion from this data set will be that both genotypes increase phosphorylation similarly, and this conclusion now fits with the lack of difference between genotypes in the Akt data. Much of the paper’s storyline is based on this data, which appears to me to be pretty shaky. Can the authors increase the n-size here?
2) AMPK activation under Igf1 overexpression is a major story in this paper. The authors only look at p-AMPK and p-TBC1D1. AMPK phosphorylation change with contraction was not different between genotypes, while TBC1D1 phosphorylation status was only elevated in the mIgf1 mice, not WT. It would be good to add an additional AMPK target (phospho-ACC or phospho-raptor) to clarify the effect of the overexpression on AMPK activity.
Minor concerns:
1) Introduction, line 65. The aim stated here is slightly different than the one stated in the abstract. It would be good to reconcile them.
2) Introduction, line 56. A reference would be helpful here (showing phosphorylation of AMPK at S485 is inhibitory).
3) Results, line 76. The use of the word “trend” here suggests the lack of a significant difference. I would revise the wording.
4) Results, line 82. I would be careful assuming that glucose uptake was necessarily different in the ITT. Blood glucose levels are determined by uptake and output (e.g., by the liver). The lower glucose levels shown in Figure 2D occurred later in the timecourse of the assay (in Figure 2C), which suggests that they could have been influenced by uptake (by muscle) or by liver output.
Author Response
Reviewer 1.
This manuscript from Chou and Barton examines the effect of genetic IGF-1 overexpression in skeletal muscle on muscle function and signaling pathways with an emphasis on AMP-activated protein kinase (AMPK). The authors conclude that the phosphorylation of IGF-1 after muscle contraction is dependent on IGF-1 overexpression. Although this should, theoretically, work to decrease AMPK activity, no indications of such a deficit were observed. The manuscript is generally well-written, with a straightforward experimental design, but I have several comments and concerns regarding its suitability for publication as listed below.
Major Concerns:
- I suspect that many of the experiments reported here are underpowered. For instance, in Fig. 3E, the distribution of data here strongly suggests that the number of WT animals may be insufficient to detect a difference in specific force between genotypes. Importantly, in Fig. 4C: this is a critical experiment because most of the paper’s conclusions derive from it. The phosphorylation of AMPK at S485 is on statistically increased by contraction in the IGF-1 overexpressor group. However, this appears, based on the data distribution, to be due to one seeming outlier in the mIgf1 group. Looking at the WT group, there are only five mice, and all of them increase their phosphorylation status. It is likely that with an increased WT n-size, the conclusion from this data set will be that both genotypes increase phosphorylation similarly, and this conclusion now fits with the lack of different between genotypes in the Akt data. Muscle of the paper’s storyline is based on this data, which appears to me to be pretty shaky. Can the authors increase the n-size here?
We were able to add N=2 WT and N=1 mIgf1+/+ mice that are littermates and 17 weeks old at the time of analysis. This has improved the functional assessment, as well as the evaluation of AMPK and its downstream targets. We have revised Figures 3 and 4, and we have added an additional Figure 7.
- AMPK activation under Igf1 overexpression is a major story in this paper. The authors only look at p-AMPK and p-TBC1D1. AMPK phosphorylation change with contraction was not difference between genotypes, while TBC1D1 phosphorylation status was only elevated in the mIgf1 mice, not WT. It would be good to add an additional AMPK target (phospho-ACC or phospho-raptor) to clarify the effect of the overexpression on AMPK activity.
We probed three of the original samples from each strain, as well as the additional two WT and one mIgf1+/+ with P-Raptor Ser792, and this data is now included in the revision as Figure 7. The proportional response to stimulation was similar between genotypes (~2.4 fold increase with stimulation). However, the amount of total raptor was ~3-fold higher in muscles from the mIgf1+/+ mice. This finding alters our interpretation of the results, and the discussion now includes these points.
Minor Concerns:
- Introduction, line 65. The aim stated here is slightly different than the one stated in the abstract. It would be good to reconcile them.
We have made these consistent, with more details in the aim at line 65: “The aim of this study was to determine if increased muscle IGF-I altered AMPKαSer485/491 and AMPKαThr172 phosphorylation and activity at rest or during muscle contraction.”
- Introduction, line 56. A reference would be helpful here (showing phosphorylation of AMPK at S485 is inhibitory).
References have been added.
- Results, line 76. The use of the word “trend” here suggests the lack of a significant difference. I would revise the wording.
Thank you for catching this, as the intent was a comparison to the changes in body weight. This has been changed.
- Results, line 82. I would be careful assuming that glucose uptake was necessarily different in the ITT. Blood glucose levels are determined by uptake and output (e.g., by the liver). The lower glucose levels shown in Figure 2D occurred later in the timecourse of the assay (in Figure 2C), which suggests that they could have been influenced by uptake (by muscle) or by liver output.
This is a point we hadn’t considered in our analysis of ITT. We will alter the interpretation to include the fact that we did not directly measure fluxes of glucose, only the amount. Also in response to Reviewer 2, we have included insulin levels obtained from a separate cohort of fasted mice.
Reviewer 2 Report
The manuscript by Chou and Barton sought to examine if increased muscle IGF1 would modulate muscle AMPKa and AKT phosphorylation during fatiguing exercise. They studied transgenic animals with overexpression of IGF1 in skeletal muscle. The transgenic animals had better glucose tolerance; they also exhibited better insulin tolerance. In experiments with isolated muscle, phosphorylation of AKT, AMPK and TBC1D1 was largely not modified by igf1 overexpression in muscles subjected to fatiguing contractions, except for the phosphorylation of AMPKa on residue S485/491 which appeared to be upregulated in fatiguing muscles from only the transgenic mice.
11. Regarding the GTT and ITT, could the authors comment on mechanisms of the beneficial effects observed in the igf1 transgenics? Does the insulin level differ in these animals compared to the wildtype?
22. Activation of AMPK under a condition when AKT too is activated (going by the phosphorylation of these proteins and of their substrates) is quite intriguing. Metabolically speaking, AMPK and AKT are activated under opposing metabolic conditions (for example fed state vs energy-deprivation state), as the authors mentioned in the introduction section. Thus, does the activation of both under the same set of conditions have physiological implications? For example, are there differences in muscle glucose transport?
33. Abstract L23-24: there is no data on glucose uptake in resting muscle. How did the authors reach the conclusion that insulin sensitivity was better in the igf1 transgenics?
44. Check sentence structure and tense, for example L51, L108, 114 (‘no significance’), 179-180, 251-252,255-256, and in a few other places.
55. L54: reference/s missing for the statement ‘…Ser485/491 is another AMPKα epitope that can be phosphorylated…’ Also missing reference/s for L56 ‘…Ser485/491 was identified in the past decade and is thought to inhibit AMPKα activity…’
66. L92, Fig 1: the symbol *** was not used in this figure. Check for similar corrections in Fig 3
77. L96: Fig 2: no body composition data in this figure
88. Fig 4: the difference in AMPKa S485/491 phosphorylation in response to electrical stimulation between WT and the transgenic mice appear to be driven by the response in a single animal. Have the authors considered removing this mouse from their analyses? Further, in the absence of significant effects of genotype (Fig 4E), how does one interpret the data in Fig 4C? This appears to be a main conclusion of the manuscript (see Abstract, L24-25).
99. L233-237: It is not clear how this section connects with the discussion.
110.L297, 305: were these tests (GTT, ITT) conducted in the same animals, and if yes, what interval elapsed between the two tests?
111.L344: BCA had not been mentioned before.
There are some minor edits that can improve clarity/readability, as indicated above in the comments to the authors.
Author Response
Reviewer 2.
The manuscript by Chou and Barton sought to examine if increased muscle IGF1 would modulate muscle AMPKa and AKT phosphorylation during fatiguing exercise. They studied transgenic animals with overexpression of IGF1 in skeletal muscle. The transgenic animals had better glucose tolerance; they also exhibited better insulin tolerance. In experiments with isolated muscle, phosphorylation of AKT, AMPK and TBC1D1 was largely not modified by igf1 overexpression in muscles subjected to fatiguing contractions, except for the phosphorylation of AMPKa on residue S485/491 which appeared to be upregulated in fatiguing muscles from only the transgenic mice.
- Regarding the GTT and ITT, could the authors comment on mechanisms of the beneficial effects observed in the igf1 transgenics? Does the insulin level differ in these animals compared to the wildtype?
There are two potential mechanisms for improved glucose handling in the Igf1 transgenic mice. First, given the increased muscle mass, there is a greater ability to take up glucose, simply by mass action. However, there is also the ability for IGF-I to activate hybrid receptors on the muscle membrane. This receptor has a greater affinity for IGF-I than for insulin, yet acts more like an insulin receptor than an IGF-I receptor. We have altered the test to reflect this. In addition, we have now included insulin levels in a separate cohort of fasted mice, and in our hands, we did not observe a difference.
- Activation of AMPK under a condition when AKT too is activated (going by the phosphorylation of these proteins and of their substrates) is quite intriguing. Metabolically speaking, AMPK and AKT are activated under opposing metabolic conditions (for example fed state vs energy-deprivation state), as the authors mentioned in the introduction section. Thus, does the activation of both under the same set of conditions have physiological implications? For example, are there differences in muscle glucose transport?
This is the central puzzle we were trying to address, as for both pathways, there is a need for fuel, yet these pathways are at odds with one another with respect to driving growth. While we did not determine glucose transport in the current study, the suggestion of adding analysis of raptor opens up a possible explanation for a way that high IGF-I may afford glucose uptake to proceed, and also allow for pro-growth actions to occur.
- Abstract L23-24: there is no data on glucose uptake in resting muscle. How did the authors reach the conclusion that insulin sensitivity was better in the igf1 transgenics?
The statement has been revised to reflect the data acquired.
- Check sentence structure and tense, for example L51, L108, 114 (‘no significance’), 179-180, 251-252,255-256, and in a few other places.
We have gone through the manuscript to revise the grammar and to ensure there is consistency in tense.
- L54: reference/s missing for the statement ‘…Ser485/491 is another AMPKα epitope that can be phosphorylated…’ Also missing reference/s for L56 ‘…Ser485/491 was identified in the past decade and is thought to inhibit AMPKα activity…’
These have been included.
- L92, Fig 1: the symbol *** was not used in this figure. Check for similar corrections in Fig 3
These have been corrected.
- L96: Fig 2: no body composition data in this figure.
This has been removed.
- Fig 4: the difference in AMPKa S485/491 phosphorylation in response to electrical stimulation between WT and the transgenic mice appear to be driven by the response in a single animal. Have the authors considered removing this mouse from their analyses? Further, in the absence of significant effects of genotype (Fig 4E), how does one interpret the data in Fig 4C? This appears to be a main conclusion of the manuscript (see Abstract, L24-25).
We have added more N’s to the WT and Tg groups for blotting analysis for Figure 4.
- L233-237: It is not clear how this section connects with the discussion.
This section has been revised, and is now lines 561-582.
- L297, 305: were these tests (GTT, ITT) conducted in the same animals, and if yes, what interval elapsed between the two tests?
These were performed on the same animals with at least 48 hours between tests. This has now been clarified in the methods.
- L344: BCA had not been mentioned before.
This has been spelled out.
Comments on the Quality of English Language
There are some minor edits that can improve clarity/readability, as indicated above in the comments to the authors.
We have gone through the text and revised to improve clarity and readability.
Round 2
Reviewer 1 Report
The authors have adequately addressed my concerns.
Author Response
Thank you for your positive statement.
Reviewer 2 Report
The authors have made sincere efforts to address the concerns raised in the first review.
A few comments remain:
1. The manuscript was a bit challenging to review because of the inserted comments that were not cleaned up; also, in Figs (for example, Fig 1-3), there are deleted figs that should have been cleaned up. It was difficult to know which of the 2 sets in each figure was the current version. For those reasons, it was challenging to verify if the edits requested were all completed.
2. In Figure 2 and its legends (L106, and other places), it was not clear why the N changed between this and the previous versions, nor why some of the figs were replaced. Other data or n values were also changed (for example, L121-123, with not much explanation as to why.
3. In L143-147 and Fig4, it was not clear which of the 2 sets of figures the authors had in mind. In Fig 4C, the effect of contraction on S485/491 AMPK is significant only in the transgenic mice. Also, **** is not used. Unless I was looking at the wrong set of figures!
In the legends to Fig 5, 5C and D are mixed.
4. The sentence in L224-226 is awkward and appears to contradict the previous sentence where the authors mentioned a significant effect of genotype, even if the difference was not big.
L343-344 reads ‘…but it has been challenging to show similar effects in whole animals or in isolated muscles ‘. Where is the reference for this?
Also, reference is missing for L346-7: ‘Further, this increase may have entered the tissue from the circulation, where IGF-I is also abundant’
Check Sentence structure/editing for readability: L25, 81, 162-163, 169-170, 199-201, 253-258, 265-266, 310-312
In L420, did the authors mean ‘randomly fed mice’?
See comments above: there are sections where clarity/readability could be improved.
